# A Least Squares Ensemble Model Based on Regularization and Augmentation Strategy

Peng Zhang, Shuyou Zhang *, Xiaojian Liu *, Lemiao Qiu and Guodong Yi

State Key Laboratory of Fluid Power and Mechatronic Systems, Zhejiang University, Hangzhou 310027, China; absent1353@163.com (P.Z.); qiulm@zju.edu.cn (L.Q.); ygd@zju.edu.cn (G.Y.)

* Correspondence: zsy@zju.edu.cn (S.Z.); liuxj@zju.edu.cn (X.L.);
  Tel.: +86-1390-581-3010 (S.Z.); +86-1398-987-2171 (X.L.)

**Abstract:** Surrogate models are often used as alternatives to considerably reduce the computational burden of the expensive computer simulations that are required for engineering designs. The development of surrogate models for complex relationships between the parameters often requires the modeling of high-dimensional functions with limited information, and it is challenging to choose an effective surrogate model over the unknown design space. To this end, the ensemble models—combined with different surrogate models—offer effective solutions. This paper presents a new ensemble model based on the least squares method, which is a regularization strategy and an augmentation strategy; we call the model the regularized least squares ensemble model (RLS-EM). Three individual surrogate models—Kriging, radial basis function, and support vector regression—are used to compose the RLS-EM. Further, the weight factors are estimated by the least squares method without using the global or local error metrics, which are used in most existing methods. To solve the collinearity in the least squares calculation process, a regularization strategy and an augmentation strategy are developed. The two strategies help explore the unknown regions and improve the accuracy on one hand; on the other hand, the collinearity can be reduced, and the overfitting phenomenon that may occur can be avoided. Six numerical functions, from two-dimensional to 12-dimensional, and a computer numerical control (CNC) milling machine bed design problem are used to verify the proposed method. The results of the numerical examples show that RLS-EM saves a considerable amount of computation time while ensuring the same level of robustness and accuracy compared with other ensemble models. The RLS-EM used for the CNC milling machine bed design problem also shows good accuracy characteristics compared with other ensemble methods.

**Keywords:** ensemble model; regularized least squares; computer simulations; augmentation strategy

## 1. Introduction

Computational simulations, such as finite element analysis (FEA) or computational fluid dynamics, have been displaying steady progress in describing engineering systems, and these simulations play a key role in optimizing the design of complex engineering equipment. However, computer simulations may consume a considerable amount of time for complicated simulations in engineering design. Therefore, a surrogate modeling method has been developed rapidly over the last three decades as an alternative for computationally expensive simulations that consumes less time [1]. A wide variety of surrogate models have been used in engineering design, such as polynomial response surface (PRS) [2,3], Kriging (KRG) [3–6], radial basis function (RBF) [7,8], and support vector regression (SVR) [8–13]. The PRS and SVR models can identify global trends for a given input data set; whereas, owing to the interpolation characteristics, KRG and RBF have higher local accuracy around the training points. Reviews about the surrogate models can be found in [14–17].

The rapid development of various surrogate models provides researchers with a lot more flexibility while they are selecting models for engineering design problems. However, it is challenging to choose the optimal model for a specific application before all the different surrogate models are constructed [18]. Since practical engineering applications often exhibit different linear and nonlinear properties, no single surrogate model can exhibit high performance in all scenarios; each surrogate model has advantages and disadvantages [19–22]. The PRS model fits the linear relationship between the inputs and the outputs well, while the KRG and RBF model are more suited for complex nonlinear relationships between the input and output datasets. The SVR model is suitable for both linear and nonlinear relationships between the inputs and the outputs, as it can choose different kernel functions and hyper-parameters [23–30]. To make better use of the advantages of each model, as well as avoid wasting existing individual models, researchers have combined different surrogate models into an ensemble model to develop weighted average surrogate (WAS) models [31,32].

Several ensemble models have been developed in the literature, and studies have shown that the ensemble model combines the predictive power of each individual surrogate model to improve accuracy and robustness. Existing ensemble models are most commonly based on error correlation or prediction variance, and they can be classified according to global and local error metrics [31–34]. Zerpa et al. [33] constructed a WAS using several individual surrogate models for the optimization of alkaline–surfactant–polymer flooding processes, and found that the WAS exhibited better performance than individual surrogates. Goel et al. [31] proposed different approaches in which the weights of the ensemble were determined based on the generalized mean square cross-validation error (GMSE). Acar et al. [32] proposed an optimization method to calculate the weight factors by minimizing the GMSE. Zhou et al. [34] used a recursive process to obtain the weight factors, updating them in each iteration until the convergence goal was reached. The study described above shows that the weight factors are evaluated as a global metric. Unlike the global error metric, the local error metric method assesses weight factors in relatively small spaces, even point-by-point [35–39]. Acar [35] used various local error measures to construct an ensemble model and presented a local error measure of pointwise cross-validation error. Lee et al. [36] presented a v-nearest points cross-validation method to calculate the weight factors in a local region. Hierarchical design space reduction [37,40], hybrid, and adaptive meta-modeling [38] are also effective methods based on local error metrics.

The essence of an ensemble model is to assign weight factors to the known individual surrogate models and then sum the results of each model. The accuracy of each model affects the accuracy of the ensemble model; thus far, individual surrogate models are expected to have relatively high accuracy to meet the robustness and the predictive performance requirements. When high-precision individual models are obtained, the weight factors can be calculated by regression methods to improve computational efficiency and save computational cost, instead of other optimization algorithms or error metrics.

Motivated by the regression idea, this paper proposes a novel ensemble modeling technique named the regularized least squares ensemble model (RLS-EM). Three individual surrogate models, KRG, RBF, and SVR, are used to develop the RLS-EM; the least squares algorithm with a regularization strategy and an augmentation strategy is used to calculate the weight factors. The regularization strategy and augmentation strategy help to solve the collinearity problem caused by the inherent interpolation properties of the KRG and RBF models, and the similar prediction values at some sample points. The augmentation strategy is carried out on the unexplored regions, which can help improve the accuracy of the surrogate models, while the regularization strategy helps to avoid the potential overfitting phenomenon. The RLS-EM aims to take advantage of the well-performing ensemble surrogate model to guarantee the robustness and accuracy for different problems from low to relatively high dimensions. The RLS-EM aims to take advantage of the well-performing ensemble surrogate model to guarantee the robustness and accuracy for different problems from low to relatively high dimensions.

The remainder of this paper is organized as follows. In the next section, a brief introduction to the ensemble methods is presented. Then, the development of the proposed RLS-EM is described.

Several numerical functions and an engineering application are tested in the following section. Finally, the conclusions are presented.

## 2. Background of Ensemble Methods

Usually, the surrogate model technique is utilized to construct several different surrogates and select the best one. However, this scenario has two major shortcomings [41]. It is wasteful to discard the so-called inaccurate models, and the accuracy of the surrogate model is affected by the sample points. This is because the surrogate model may exhibit different precisions for different data sets. To overcome these drawbacks, ensemble methods are proposed.

An ensemble surrogate model is a weighted combination of several individual surrogate models [42], which is defined as:

$$\hat{f}_e(x) = \sum_{i=1}^{M} w_i(x)\hat{f}_i(x), \quad \sum_{i=1}^{M} w_i(x) = 1 \tag{1}$$

where $\hat{f}_e(x)$ is the prediction of the ensemble, $M$ is the number of surrogate models used, and $w_i$ is the weight factor for the $i$th surrogate $\hat{f}_i(x)$. Evidently, the larger weights are assigned to the more accurate surrogate models, and vice versa.

Zerpa et al. [33] proposed the evaluation of the weight factors $w_i$ in a linear ensemble as:

$$w_i = \frac{1}{V_i} / \sum_{j=1}^{M} \frac{1}{V_j} \tag{2}$$

where $V_i$ is the prediction variance of the $i$th surrogate model.

Goel et al. [31] considered PRS, KRG, and RBF, and proposed an ensemble scheme to estimate the weight factors in a WAS, including the BestPRESS (BP), the PRESS weighted surrogate (PWS), and the non-parametric PRESS weighted surrogate (NPWS).

Taking the prediction sum of squares (PRESS) as the error measure, the NPWS is given as:

$$w_i = \sum_{j=1,j\neq i}^{M} E_i / \left[ (M-1) \sum_{j=1}^{M} E_j \right] \tag{3}$$

where $E_j$ is the GMSE of the $i$th surrogate model calculated from:

$$E_j = \frac{1}{N} \sum_{i=1}^{N} \left[ y(x_i) - \hat{y}_j^{(-i)}(x_i) \right]^2 \tag{4}$$

where $y(x_i)$ is the true response at the $i$th data point $x_i$, and $\hat{y}_j^{(-i)}(x_i)$ is the corresponding prediction from the surrogate model constructed using all except the $i$th data point $x_i$, and $N$ is the number of sample points.

The model with the least PRESS error is assigned a weight factor of one, and all the other models are assigned zero weight factors; this strategy is called the BP model [31].

The PWS uses the GMSE as a global error metric to select the weight factors using a heuristic formulation, which is formulated as follows:

$$
\begin{aligned}
w_i &= w_i^* / \sum_{j=1}^{M} w_j^* \\
w_i^* &= (E_i + \alpha E_{avg})^\beta, \ \beta < 0, \ \alpha < 1 \\
E_{avg} &= \frac{1}{M} \sum_{i=1}^{M} E_i
\end{aligned}
\tag{5}
$$

The weighting scheme requires the user to specify parameters $\alpha$ and $\beta$, which control the contribution of the individual surrogates; $\alpha$ and $\beta$ are assumed to be 0.05 and $-1$, respectively [31].

Acar and Rais-Rohani [32] used GMSE as the global error metric and proposed an optimization algorithm to calculate the weight factors; the algorithm is expressed as follows:

$$\begin{aligned} & \text{Find } w_i \\ & \text{Min } GMSE \\ & \text{s.t.} \quad \sum_{i=1}^{M} w_i = 1 \end{aligned} \tag{6}$$

Viana et al. [43] proposed an ensemble surrogate model called optimal weighted surrogate (OWS); the OWS is represented as follows:

$$\begin{aligned} & \min_{w_i} MSE(\hat{f}_e(x)) = w^{\mathrm{T}} C w \\ & \text{s.t.} \quad \sum_{j=1}^{M} w_j = 1 \end{aligned} \tag{7}$$

The correlation matrix of the error from the individual surrogate models that are used to constitute the ensemble surrogate model is expressed as follows:

$$C_{ij} = \frac{1}{N} e_i^{\mathrm{T}} e_j \tag{8}$$

where $e_i$ and $e_j$ are the vectors of cross-validation errors (i.e., PRESS) for the $i$th and $j$th surrogate models, respectively. The application of the ensemble models can be found in [44–47].

## 3. Proposed Regularized Least Squares Ensemble Model

### 3.1. Basic Formulation of the Least Squares Method

A general linear regression model can be represented as follows [47]:

$$y = w_0 + \sum_{i=1}^{M} w_i p_i(x) + \varepsilon \tag{9}$$

where $p_i(x)$ represents any function about the variable $x$ or simply the variable $x$, $M$ is the number of regression terms, and $\varepsilon$ is the approximation error. For convenience, $p_i(x)$ is characterized with $X_i$, for $N$ samples with $M$ dimensions, $X = [p_1(x_i), p_2(x_i), \ldots, p_M(x_i)]$, and the corresponding responses $Y = [y_1, y_2, \ldots, y_N]^{\mathrm{T}}$. Then, the matrix form of linear regression is represented as:

$$Y = Xw + \varepsilon \tag{10}$$

where $w = [w_1, w_2, \ldots, w_M]^{\mathrm{T}}$, and the error term $\varepsilon = [\varepsilon_1, \varepsilon_2, \ldots, \varepsilon_N]^{\mathrm{T}}$, supposing that the errors are normally and independently distributed, with zero mean and finite variance, that is $\varepsilon_i \sim N(0, \sigma^2)$. Based on the Gauss–Markov theorem [47], the weight factors $\hat{w} = [\hat{w}_1, \hat{w}_2, \cdots, \hat{w}_M]$ calculated by the OLS method form the best linear unbiased estimator, which can be represented as:

$$\hat{w} = \left(X^{\mathrm{T}} X\right)^{-1} X^{\mathrm{T}} Y \tag{11}$$

and $\hat{w}$ satisfies the following equations:

$$
\begin{aligned}
E(\hat{w}) &= E(w) \\
Var(\hat{w}) &= \sigma^2 (X^T X)^{-1} \\
\sigma^2 &= Var(\varepsilon_1) = Var(\varepsilon_2) = \cdots Var(\varepsilon_N)
\end{aligned}
\tag{12}
$$

### 3.2. Samples Adding by the Augmentation Strategy

The RLS-EM proposed in this paper seeks to simultaneously capture the global and local accuracy. Since the PRS may exhibit lower accuracy in some nonlinear applications, RLS-EM only combines KRG, RBF, and SVR to meet the local accuracy and global trend requirements. The predicted values of the KRG and RBF models at the training points are equal to the actual function values, so the collinearity is unavoidable. To solve the collinearity problem, an augmentation strategy and a regularization strategy were developed. The augmentation strategy is used to reduce the influence of the collinearity on one hand; on the other hand, the augmentation strategy helps to improve the accuracy of the model in the unexplored area.

The $N$ samples obtained by Latin hypercube sampling (LHS) technique are used to construct the KRG, RBF, and SVR surrogate models, and the corresponding prediction values at the samples are $\hat{Y} = [\hat{y}_1(x_i), \hat{y}_2(x_i), \hat{y}_3(x_i)]$, $i = 1, 2, \ldots, N$, $\hat{y}_1(x_i), \hat{y}_2(x_i), \hat{y}_3(x_i)$ represents the KRG, RBF, and SVR prediction values at $x_i$, respectively, the corresponding actual function values $Y = [y_1, y_2, \ldots, y_N]^T$. The augmentation strategy is implemented to add additional samples in the exploration regions that are far from the $N$ original samples. The number of $N_{add}$ points (the set is $X_{add}$) is obtained from Algorithm 1.

---

**Algorithm 1** Pseudo code of augmentation strategy for adding samples

---

**Input**: $X = [x_1, x_2, \ldots, x_N]$.
1: Set $S_{add}^{pri}$ empty, $S = X$.
2: Obtain $3 \times N_{add}$ samples by LHS, put them in $X_{lhs}$.
3: For $i = 1: 3N_{add}$ do
4: Calculate the distance of all the members in $X_{lhs}$ to the samples in $S$.
5: Move the sample with the largest distance from $X_{lhs}$ to $S_{add}^{pri}$ and $S$.
6: End for
7: Construct KRG, RBF by $S$, calculate the uncertainties with (13) at the sample set $S_{add}^{pri}$, storage the difference values in $P_{kr}$.
8: Sort $P_{kr}$ from the largest to the least, choose the top $N_{add}$ values of the corresponding samples, and put them into $X_{add}$.
**Output**: $X_{add}$.

---

$$
P_{kr}(x_i) = \left| \hat{y}_1(x_i) - \hat{y}_2(x_i) \right|
\tag{13}
$$

where the $P_{kr}(x_i)$ is the absolute value of the difference between the KRG and RBF model at the $i$th adding points. The prediction values of the KRG, RBF, and SVR surrogate models on $X_{add}$ are represented as $\hat{Y}_{add} = [\hat{y}_1(x_{iadd}), \hat{y}_2(x_{iadd}), \hat{y}_3(x_{iadd})]$, $iadd = 1, 2, \ldots, N_{add}$, and the actual functions values at the $N_{add}$ points are $Y = [y_{N+1}, y_{N+2}, \ldots, y_{N+Nadd}]^T$. Then, the augmentation matrix system of $\hat{Y}_{expand}$ and $Y$ are represented as:

$$\hat{Y}_{expand} = \begin{bmatrix} \hat{y}_1(x_1) & \hat{y}_2(x_1) & \hat{y}_3(x_1) \\ \vdots & \vdots & \vdots \\ \hat{y}_1(x_N) & \hat{y}_2(x_N) & \hat{y}_3(x_N) \\ \hat{y}_1(x_{N+1}) & \hat{y}_2(x_{N+1}) & \hat{y}_3(x_{N+1}) \\ \vdots & \vdots & \vdots \\ \hat{y}_1(x_{N+N_{add}}) & \hat{y}_2(x_{N+N_{add}}) & \hat{y}_3(x_{N+N_{add}}) \end{bmatrix}, Y = \begin{bmatrix} y_1 \\ \vdots \\ y_N \\ y_{N+1} \\ \vdots \\ y_{N+N_{add}} \end{bmatrix} \tag{14}$$

### 3.3. The Regularization Strategy in the Least Squares System

A regularization term is added to further reduce the impact of collinearity. Due to the interpolation properties of the KRG and RBF models, $\hat{y}_1(x_i)$ and $\hat{y}_2(x_i)$ are equal to the actual function values at the $N$ samples. The relatively high precision of KRG, RBF, and SVR surrogate models may also predict approximately equal values at some samples in the set $X_{add}$. By adding a regularization term multiplying an identity matrix, the matrix coefficients $\hat{w}_{expand}$ can be estimated from the augmented matrix inversion system as follows:

$$\hat{w}_{expand} = (\hat{Y}_{expand}^{\mathrm{T}} \hat{Y}_{expand} + \lambda I)^{-1} \hat{Y}_{expand}^{\mathrm{T}} Y \tag{15}$$

where $I_{3\times3}$ is an identity matrix, and $\lambda$ is the regularization parameter. Since the linear correlation in $\hat{Y}_{expand}^{\mathrm{T}}$ is expected to be lower than that in $\hat{Y}$, and the regularization item further reduces the linear correlation, the accuracy and robustness on evaluating $w$ by means of $\hat{w}_{expand}$ is expected to be better. The weight factors for the ensemble model are calculated as:

$$w_i^* = \frac{\hat{w}_{expand}^i}{\sum\limits_{i=1}^{3} \hat{w}_{expand}^i}, i = 1, 2, 3 \tag{16}$$

The regularized least squares ensemble method can be expressed as follows:

1. Random sampling $N$ samples, the $N_{add}$ samples are obtained by the augmentation strategy, and the actual function values $Y = [y_1, y_2, \ldots, y_{N+Nadd}]$ are calculated by expensive simulations.
2. Choose $N$ samples to construct the KRG, RBF, and SVR surrogate models, as the prediction values of the KRG and RBF at the $N$ samples are equal to the corresponding actual function values of $y_i$, $i = 1, 2, \ldots, N$; calculate $\hat{y}_3(x_i)$ of the SVR model at each of the $N$ samples.
3. Evaluate $\hat{y}_1(x_{N+i})$, $\hat{y}_2(x_{N+i})$, and $\hat{y}_3(x_{N+i})$ for the KRG, RBF, and SVR surrogate models, where $i = 1, 2, \ldots, N_{add}$ and $N_{add}$ is the number of adding samples. Construct the matrix $\hat{Y}_{expand}$ and $Y$ as in Equation (14).
4. Calculate the inverse of the augmented matrix system for $\hat{w}_{expand}$ by Equation (15), and the standardized weight factors by Equation (16).

However, the regularization parameter $\lambda$ should be confirmed before using Equation (15); a search algorithm was developed to obtain the optimal regularization parameter value $\lambda^*$, and the detailed pseudo codes are summarized in Algorithm 2.

---

**Algorithm 2** Search for the optimal regularization parameter $\lambda^*$

---

**Begin**:

1: A constant array $\prod = \{\xi_1, \xi_2, \cdots, \xi_q\}(0 < \xi_1 < \xi_2 < \cdots < \xi_q < 1)$ is set for $\lambda$, and $l = \min = 1$, $r = \max = q$.

2: While $\lambda_{\min} < \lambda_{\max}$, mid $= \lfloor (l+r)/2 \rfloor$, go to step 3, else go to step 8.

3: Randomly divide the predicted values of the KRG, RBF, and SVR surrogate models of the $N + N_{add}$ samples into $k$ (we use $k = 5$ in this paper) equal parts.

4: The $\hat{Y}_{train}$ matrix is made up by the predicted values of three individual surrogate models in the $k - 1$ group, and by singular value decomposition (SVD), which can be expressed as

$$\hat{Y}_{train} = UDV^T \tag{17}$$

where $\hat{Y}_{train} = [\hat{Y}_1, \hat{Y}_2, \hat{Y}_3]$, $U = (U_1, U_2, U_3)$ is a $m_t \times 3$ orthogonal matrix, and $m_t$ is the number of the $k-1$ parts of the samples. $D = \text{diag}(d_1, d_2, d_3)$ is a $3 \times 3$ diagonal matrix. $V = (V_1, V_2, V_3)$ is a $3 \times 3$ orthogonal matrix.

5: After the SVD, $\hat{w}_e$ is calculated for $\lambda_l$ and $\lambda_r$ by:

$$\hat{w}_e = V\text{diag}(\frac{d_j}{d_j + \lambda})U^T Y_{train}$$
$$\hat{w}_e = [\hat{w}_{e1}, \hat{w}_{e2}, \hat{w}_{e3}], j = 1, 2, 3 \tag{18}$$

$\hat{w}_1$ is the initial weight factors for $\lambda_l$, and $\hat{w}_2$ is the initial weight factors for $\lambda_r$.

6: Calculate the weight factors with Equations (15) and (16) for $\hat{w}_1$ and $\hat{w}_2$, separately, and construct the $\hat{f}_e(\lambda_l)$ and $\hat{f}_e(\lambda_r)$ with Equation (1).

7: Calculate the *RMSE* of $\hat{f}_e(\lambda_l)$ and $\hat{f}_e(\lambda_r)$ with Equation (20), and the current optimal $\lambda_c$ values are calculated as:

$$\text{if } RMSE(\hat{f}_e(\lambda_{\min})) < RMSE(\hat{f}_e(\lambda_{\max})),$$
$$\lambda_c = \lambda_{\min}, \lambda_r = \lambda_{\text{mid}}$$
$$\text{else} \tag{19}$$
$$\lambda_c = \lambda_{\max}, \lambda_l = \lambda_{\text{mid}}$$

Back to 2

8: The optimal $\lambda^*$ is equal to the $\lambda_c$ after iteration, and it can be used to construct the RLS-EM.

**Output**: $\lambda^*$.

---

## 4. Case Studies

In this section, we compare the performance of the RLS-EM with that of the individual models KRG, RBF, and SVR (the detailed construction can be seen in Appendix A) and the ensemble models BP, PWS, NPWS, and OWS described in Section 2. Three types of error metrics were used to evaluate the performances of different surrogate models: root mean squared error (*RMSE*), which provides a global error measure over the design space; average absolute error (*AAE*), which ensures that the positive and negative errors will not counteract; and the coefficient of determination ($R^2$), which is a statistical measure of how close the data are to the fitted regression line.

$$RMSE = \sqrt{\frac{1}{N_t} \sum_{i=1}^{N_t} (y_i - \hat{y}_i)^2}$$

$$AAE = \frac{\sum_{i=1}^{N_t} |y_i - \hat{y}_i|}{N_t} \tag{20}$$

$$R^2 = 1 - \frac{\sum_{i=1}^{N_t} (y_i - \hat{y}_i)^2}{\sum_{i=1}^{N_t} (y_i - \overline{y})^2}$$

where $\overline{y}$ is the mean of the observed responses, $y_i$ denotes the observed response for $x_i$, $\hat{y}_i$ denotes the corresponding prediction, and $N_t$ is the number of evaluation points.

We implement the RLS-EM with MATLAB routines, the KRG model was based on a design and analysis of computer experiment toolbox named DACE [48], the RBF model was developed by

Sarra [49], and the SVR model was based on the LIBSVM, a library for support vector machines, which was developed by Chang and Lin [50]. Four ensemble models including BP, PWS, NPWS, and OWS were implemented in the MATLAB toolbox developed by Viana [51]. The cases have been executed with MATLAB R2018a on a computer Intel (R) Core (TM) i7-8700K, CPU @3.7 GHz, 32.0 Gb RAM, 64 bits, and Windows 10.

*4.1. Numerical Examples*

Six numerical examples varying from two-dimensional (2-D) to 12-dimensional (12-D) [42,44] were chosen to test the performance of RLS-EM: (1) Branin-Hoo function; (2) Camelback function; (3) Hartmann-3 function; (4) Hartmann-6 function; (5) extended Rosenbrock function (9-D); and (6) Dixon–Price function (12-D). A description of each test is given in Appendix B.

LHS was used to generate the training and testing sets, the MATLAB routine "lhsdesign" with "maximin" criterion and 100 iterations were used to generate the ($N + N_{add}$) samples and $N_t$ tests. The summary of the sampling in the numerical cases is provided in Table 1.

**Table 1.** Summary specifications for numerical cases.

| Function | $ndv$ [1] | $N$ | $N_{add}$ | $N_t$ |
|---|---|---|---|---|
| Branin-Hoo | 2 | 20 | 6 | 400 |
| Camelback | 2 | 20 | 6 | 400 |
| Hartman-3 | 3 | 30 | 9 | 1000 |
| Hartman-6 | 6 | 100 | 30 | 1000 |
| Extended Rosenbrock | 9 | 150 | 45 | 1000 |
| Dixon–Price | 10 | 200 | 60 | 1000 |

[1] $ndv$: Variable dimension, $N_{add}$: 30% of $N$ [41].

Table 2 lists the setup details of the individual models, which were used to develop the ensemble model based on different variable dimensions and nonlinearities. Each individual model has significant differences between variables with different dimensions and different degrees of nonlinearity, e.g., for the low-dimensional variables such as variable with numbers two and four, constant regression can satisfy the accuracy requirements of a KRG model, while the high-dimensional variables require quadratic regression to obtain a more accurate model. Similarly, the kernel parameters and regularization parameters of different dimensional variables with different degrees of nonlinearity are different for the SVR model. Thus, the KRG, RBF, and SVR model setting information for different dimensional variables are listed in detail, as shown in Table 2.

To validate the performance of the different surrogate models, 100 runs were executed for each of the numerical examples. The MATLAB routine "boxplot" was used for easy visualization and comparison. The three-dimensional surface plots of the Branin-Hoo and Camelback functions are shown in Figures 1 and 2, respectively. The nine surface plots in Figure 1 show that each surrogate model fits the Branin-Hoo function well. However, Figure 2 shows that the different surrogate models have considerable differences in the Camelback function fitting. Despite the two functions being highly nonlinear, the RLS-EM can accurately approximate the actual functions. The boxplots of RMSE, AAE, and $R^2$ for the different test functions are shown in Figures 3–5; the mean and standard deviations of the different surrogate models for the performances are listed in Table 3. After 100 runs were executed, the mean and standard deviation of the RMSE, AAE, and $R^2$ metrics for the numerical examples are shown in Table 3. For each metric of the numerical examples, the values to the left of the symbol "/" are the mean of the different models, and the values below are the standard deviations corresponding to the models. For the RMSE and AAE metrics, the smaller mean values indicate the better model accuracy, and the smaller standard deviation values show the better robustness. The $R^2$ metric with a mean value is closer to one and a smaller standard deviation indicate a more accurate and more robust model.

**Table 2.** Surrogate models setup details. KRG: Kriging, RBF: radial basis function, SVR: support vector regression.

| *ndv* | Model | Details [1] |
|---|---|---|
| 2 | KRG | Constant regression, Gaussian correlation, $\theta_0 = ndv^{(1/2)}$, $0.01 < \theta_i < 20$ |
|  | RBF | Gaussian basis functions, kernel parameter $\gamma = 4$, no polynomial term |
|  | SVR | Gaussian kernel $\gamma = 5$, regularization parameter $C = \infty$, quadratic loss $\varepsilon = 0.01$ |
| 3 | KRG | Constant regression, Gaussian correlation, $\theta_0 = ndv^{(1/3)}$, $0.01 < \theta_i < 20$ |
|  | RBF | Gaussian basis functions, kernel parameter $\gamma = 0.5$, No polynomial term |
|  | SVR | Gaussian kernel $\gamma = 0.5$, regularization parameter $C = 100$, quadratic loss $\varepsilon = 0.01$ |
| 6 | KRG | Linear regression, Gaussian correlation, $\theta_0 = ndv^{(1/6)}$, $0.01 < \theta_i < 20$ |
|  | RBF | Gaussian basis functions, kernel parameter $\gamma = 0.5$, no polynomial term |
|  | SVR | Gaussian kernel $\gamma = 0.5$, regularization parameter $C = \infty$, quadratic loss $\varepsilon = 0.001$ |
| 9 | KRG | Linear regression, Gaussian correlation, $\theta_0 = ndv^{(1/9)}$, $0.01 < \theta_i < 20$ |
|  | RBF | Gaussian basis functions, kernel parameter $\gamma = 1$, polynomial term = 1 |
|  | SVR | Gaussian kernel $\gamma = 0.5$, regularization parameter $C = 100$, quadratic loss $\varepsilon = 0.001$ |
| 12 | KRG | Quadratic regression, Gaussian correlation, $\theta_0 = ndv^{(1/12)}$, $0.01 < \theta_i < 20$ |
|  | RBF | Gaussian basis functions, kernel parameter $\gamma = 2$, polynomial term = 1 |
|  | SVR | Gaussian kernel $\gamma = 0.5$, regularization parameter $C = 100$, quadratic loss $\varepsilon = 0.0001$ |

[1] The above parameters were set according to experience and can be fine-tuned by many optimization algorithms, which is not covered in this study.

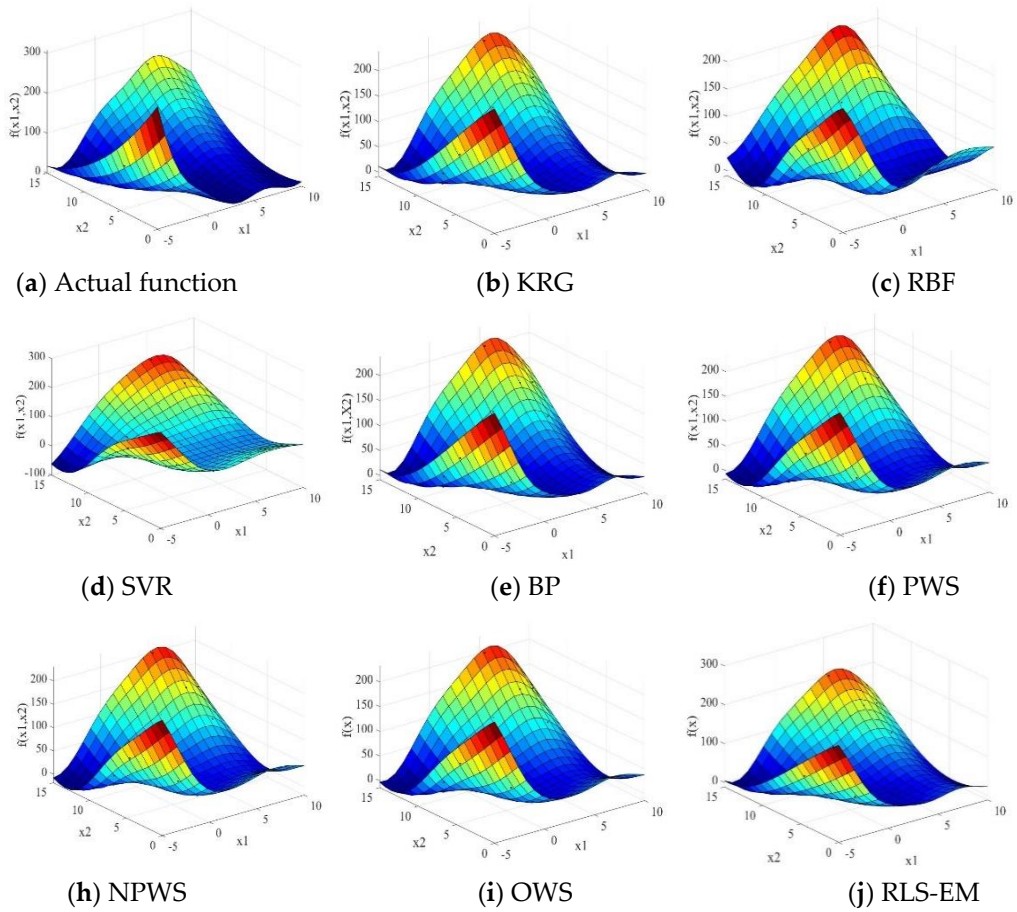

(**a**) Actual function          (**b**) KRG          (**c**) RBF

(**d**) SVR          (**e**) BP          (**f**) PWS

(**h**) NPWS          (**i**) OWS          (**j**) RLS-EM

**Figure 1.** Surface plots of the Branin-Hoo function.

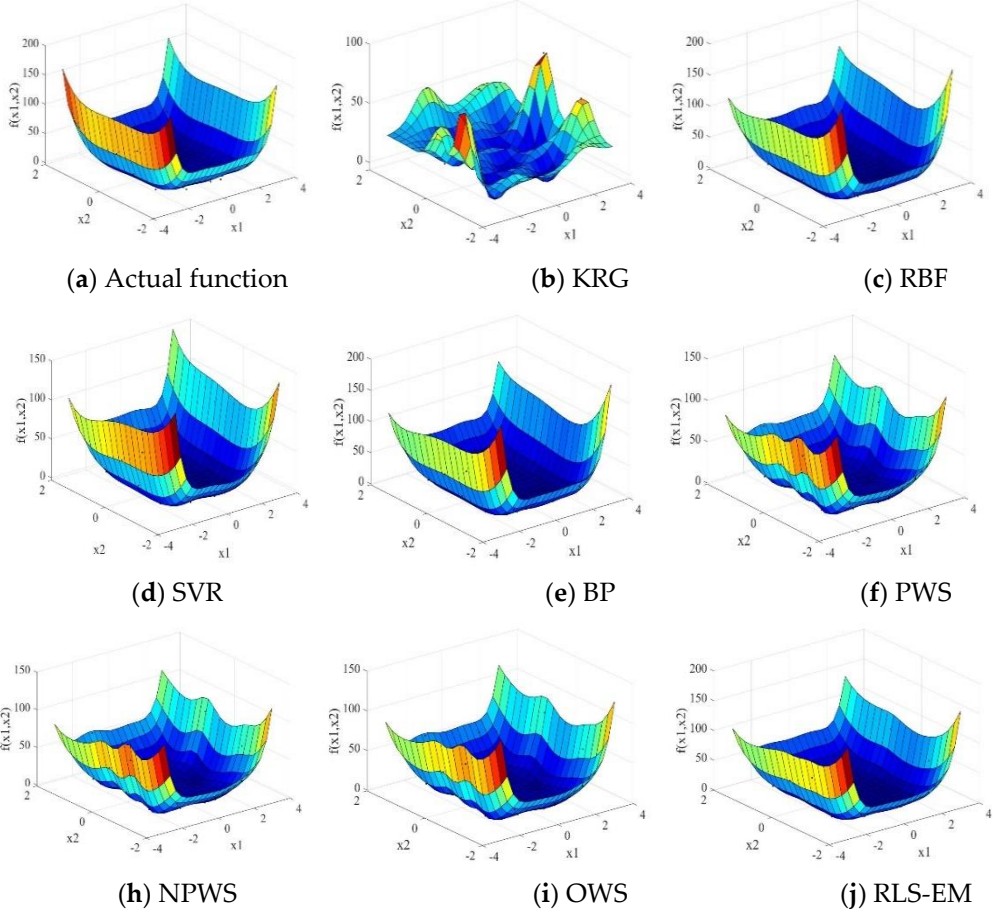

**Figure 2.** Surface plots of the Camelback function.

From Table 3, and Figures 3–5, we can see that no individual surrogate model is always accurate for all test cases, KRG fits the Branin-Hoo function well, while RBF shows a better fitting precision than KRG for the Camelback function. The superiority of ensemble models is not evident for low-dimensional variables functions; however, as the variable dimension and the degree of nonlinearity increase, the ensemble models perform better than most of the individual surrogate models. RLS-EM outperforms all the models in most of the error metrics for the six numerical problems. It shows good fitting performance and lower RMSE values on the Camelback, Hartmann-3, Hartmann-6, and Dixon–Price functions. The RMSE and AAE values in Table 3 and their boxplots in Figures 3–5 also show that the RLS-EM is robust.

**Table 3.** Comparison of root mean squared error (RMSE), average absolute error (AAE), and $R^2$ for different surrogate models. BP: BestPRESS, PRESS: prediction sum of squares, PWS: PRESS weighted surrogate, NPWS: non-parametric PRESS weighted surrogate, OWS: optimal weighted surrogate, RLS-EM: regularized least squares ensemble model.

| Function | Metric | KRG | RBF | SVR | BP | PWS | NPWS | OWS | RLS-EM |
|---|---|---|---|---|---|---|---|---|---|
| Branin-Hoo | RMSE [1] | **11.855**/4.452 | 19.127/4.461 | 18.981/4.591 | 13.869/5.920 | 15.451/4.372 | 15.582/4.333 | 15.184/4.475 | 12.008/4.679 |
| | AAE | **6.032**/1.971 | 11.278/2.020 | 11.315/2.154 | 7.577/3.312 | 8.670/1.942 | 8.771/1.902 | 8.460/2.063 | 6.436/2.317 |
| | $R^2$ | **0.941**/**0.045** | 0.859/0.068 | 0.860/0.071 | 0.917/0.068 | 0.905/0.055 | 0.904/0.055 | 0.908/0.056 | 0.939/0.050 |
| Camel back | RMSE | 19.490/4.823 | 7.855/5.755 | 13.005/4.262 | 10.961/7.590 | 11.190/3.312 | 11.441/**3.027** | 10.842/4.059 | **7.812**/3.862 |
| | AAE | 11.367/2.764 | **4.591**/2.366 | 7.136/2.147 | 6.389/3.683 | 6.519/1.676 | 6.651/**1.551** | 6.329/2.001 | 4.834/2.148 |
| | $R^2$ | 0.698/0.157 | 0.930/0.197 | 0.859/0.091 | 0.867/0.221 | 0.898/0.073 | 0.895/0.064 | 0.899/0.097 | **0.943**/**0.062** |
| Hart mann-3 | RMSE | 0.235/0.060 | 0.417/0.049 | 0.372/0.076 | 0.253/0.083 | 0.273/0.045 | 0.278/**0.044** | 0.265/0.048 | **0.233**/**0.044** |
| | AAE | 0.159/0.036 | 0.281/0.030 | 0.221/0.035 | 0.170/0.051 | 0.176/0.024 | 0.179/**0.023** | 0.171/0.027 | 0.157/0.034 |
| | $R^2$ | 0.929/0.038 | 0.788/0.049 | 0.826/0.077 | 0.914/0.061 | 0.907/0.032 | 0.905/0.032 | 0.913/0.033 | **0.931**/**0.030** |
| Hart mann-6 | RMSE | 0.239/0.034 | 0.192/0.019 | 0.214/0.025 | 0.193/0.021 | 0.196/0.023 | 0.196/0.023 | 0.195/0.023 | **0.190**/**0.019** |
| | AAE | 0.156/0.022 | 0.115/**0.008** | 0.115/**0.008** | 0.116/0.011 | 0.112/0.010 | 0.112/0.010 | **0.111**/0.009 | 0.114/**0.008** |
| | $R^2$ | 0.588/0.114 | 0.739/0.046 | 0.677/0.055 | 0.734/0.057 | 0.728/0.050 | 0.727/0.051 | 0.731/0.049 | **0.743**/**0.045** |
| Extended Rosen-brock | RMSE ($* 10^5$) | 0.201/0.023 | 0.185/0.021 | 0.201/0.027 | 0.187/0.022 | **0.180**/0.021 | 0.181/0.022 | **0.180**/0.022 | 0.184/**0.020** |
| | AAE ($* 10^5$) | 0.154/0.017 | 0.142/**0.015** | 0.154/0.019 | 0.144/0.017 | **0.138**/0.016 | **0.138**/0.016 | 0.138/0.016 | 0.141/**0.015** |
| | $R^2$ | 0.765/0.028 | 0.801/0.024 | 0.764/0.036 | 0.796/0.027 | 0.811/0.023 | 0.811/0.023 | **0.812**/**0.022** | 0.808/0.023 |
| Dixon–Price | RMSE ($* 10^6$) | 0.159/**0.017** | 0.195/0.022 | 0.230/0.029 | 0.161/0.019 | 0.175/0.020 | 0.177/0.020 | 0.171/0.019 | **0.158**/0.019 |
| | AAE ($* 10^6$) | 0.238/0.034 | **0.151**/**0.017** | 0.174/0.022 | **0.151**/0.019 | 0.166/0.020 | 0.168/0.020 | 0.162/0.019 | 0.160/0.019 |
| | $R^2$ | 0.835/**0.025** | 0.753/0.034 | 0.760/0.045 | 0.831/0.033 | 0.802/0.026 | 0.797/0.026 | 0.810/**0.025** | **0.841**/0.057 |

[1] The best error value in each category is shown in bold for ease of comparison.

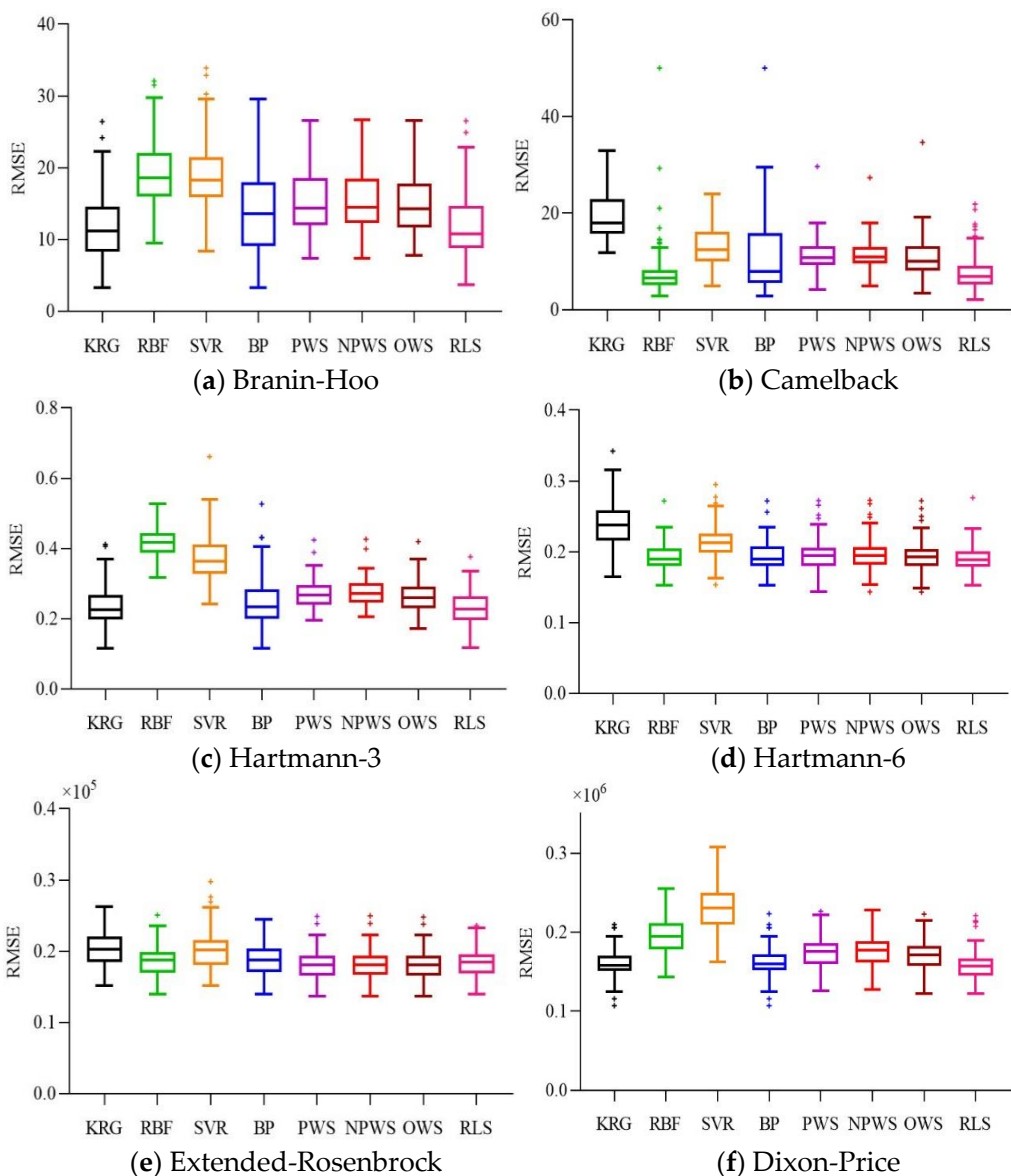

**Figure 3.** Boxplots of RMSE for the six numerical examples.

The BP, PWS, NPWS, and OWS use GMSE as error metric, and they require more computation time than the individual surrogate models, especially for high-dimensional problems. The GMSE error metric takes a relatively longer time to repeatedly construct the individual surrogate models, and the computation time is also affected by the number of divisions. In the RLS-EM, the individual surrogate models are constructed based on the initial samples, the weight factors are obtained by the regularization least squares method, which helps avoid the time spent on repetitively constructing the individual surrogate models. Figure 6 shows the computational cost of Hartmann-6, Extended-Rosenbrock, and Dixon-Price problems, which are represented by the subscript numbers of 1, 2 and 3, respectively. Further, Figure 6 shows that, as the variable dimension increases, BP, PWS, NPWS, and OWS are considerably more time-consuming than RLS-EM.

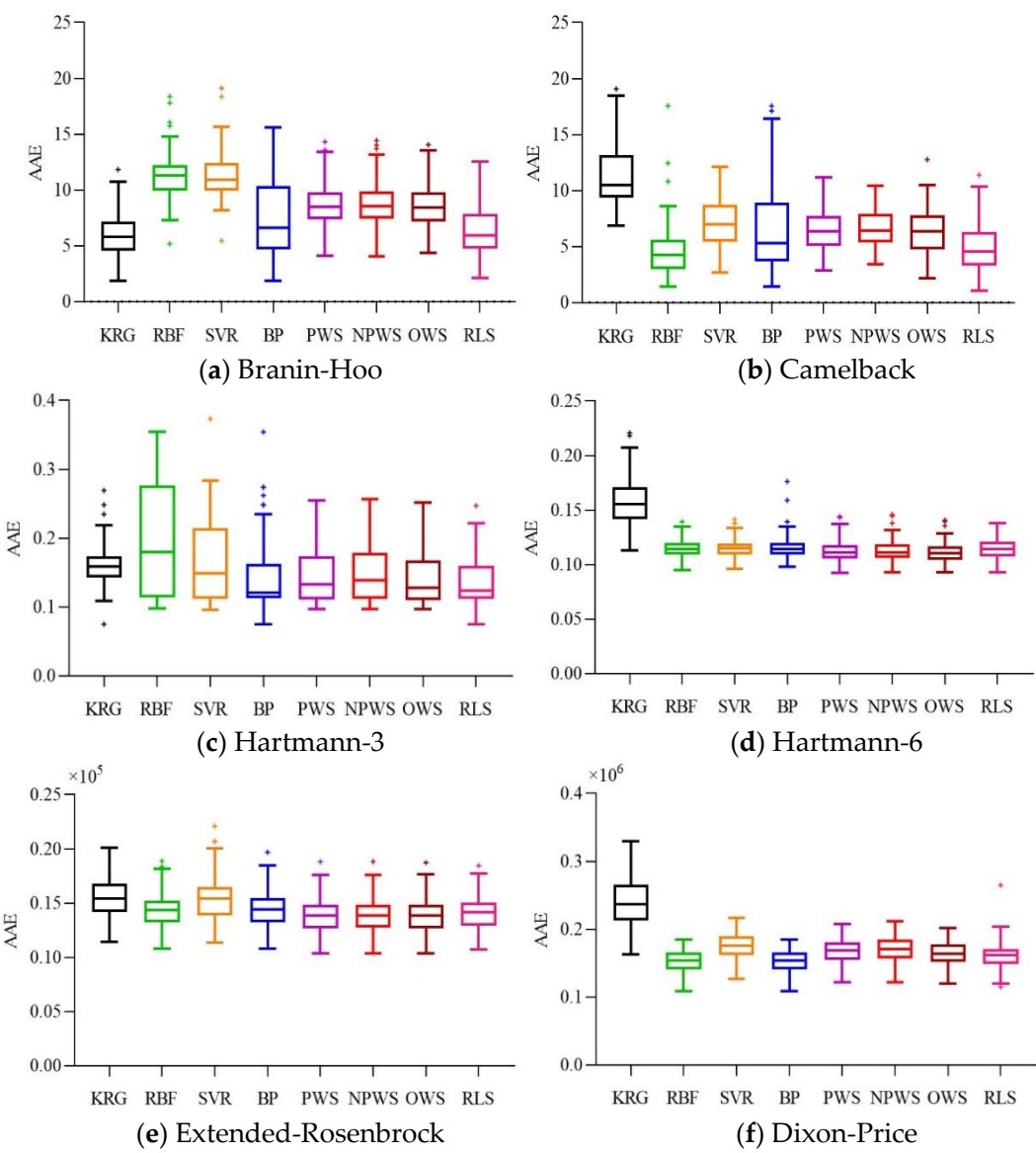

**Figure 4.** Boxplots of AAE for the six numerical examples.

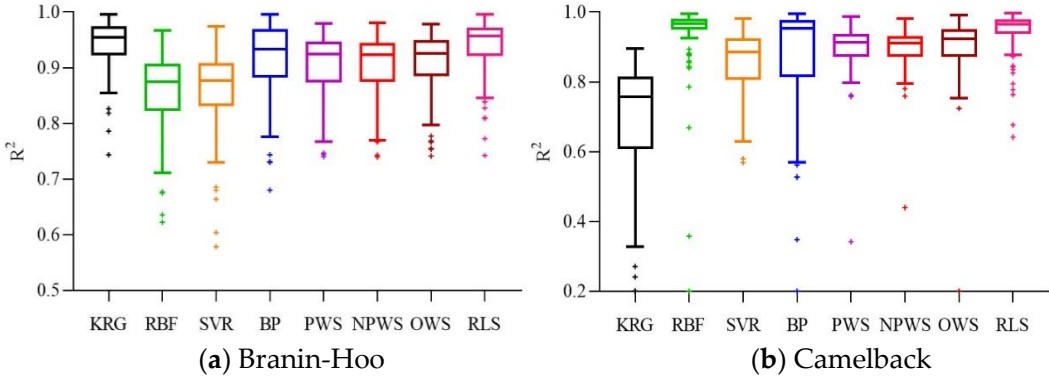

**Figure 5.** *Cont.*

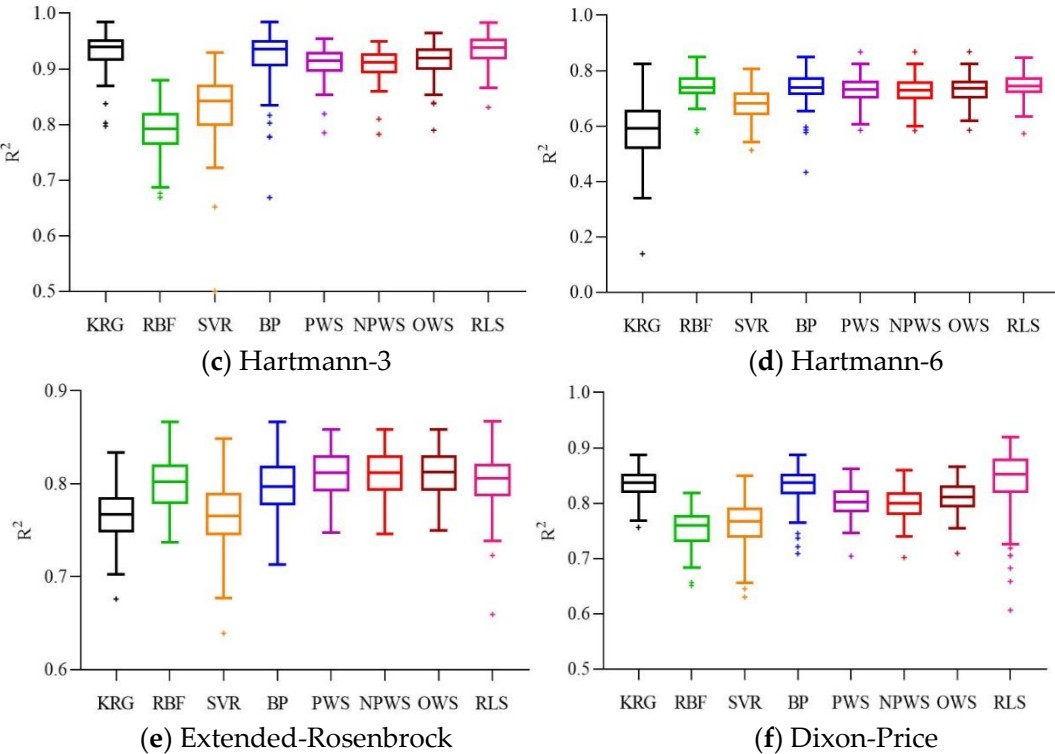

**Figure 5.** Boxplots of $R^2$ for the six numerical examples.

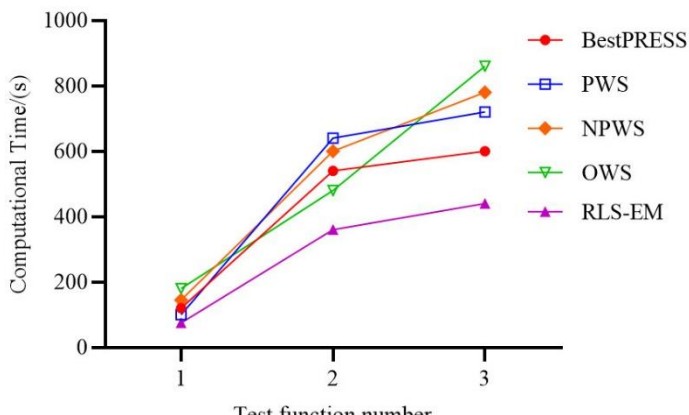

**Figure 6.** Comparison of computational time of ensemble models.

*4.2. Deformation Prediction for the CNC Milling Machine Bed*

A CNC milling machine is mainly composed of a bed, column, slider, and toolbox among other components. The column and slider, under static conditions, exert a large force on the bed, which is expressed by the red arrows in Figure 7. When the milling machine is being operated, the bed is also affected by the milling impact from the toolbox. Since the deformation has a great influence on the milling precision, the design of the milling machine bed needs good resistance to the deformation; thus, it is very important to accurately predict the deformation during design.

As the milling force is small, we only considered the column and slider weights applied to the milling machine bed, and we predicted the static deformation. The simplified structure of the bed is mainly controlled by eight variables, which are shown in Figure 8. The variables' design space is set as $x_1 \in [40,60]$, $x_2 \in [40,60]$, $x_3 \in [50,80]$, $x_4 \in [40,60]$, $x_5 \in [20,40]$, $x_6 \in [20,45]$, $x_7 \in [15,30]$, $x_8 \in [50,80]$, and $x_9 \in [40,60]$; all the variables are in millimeters. The force of the beam is 56.5 kN and that of the slider is 23.68 kN. The slider is positioned at the initial position of the bed. FEA simulations were

carried out to obtain the sample set and the corresponding deformation values. An RLS-EM model was constructed to evaluate the deformation of the bed under the two forces, which are based on the variables with different size values. A total number of 200 sample points were selected for the construction and verification of the proposed ensemble model.

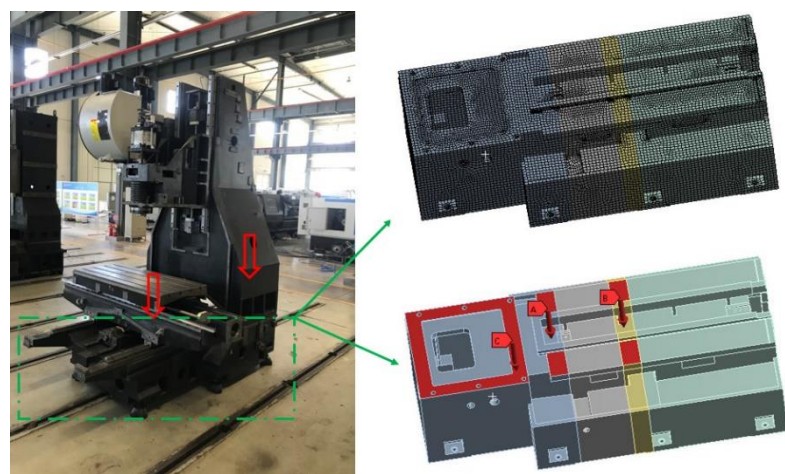

**Figure 7.** Milling machine tool and the sketch of the bed.

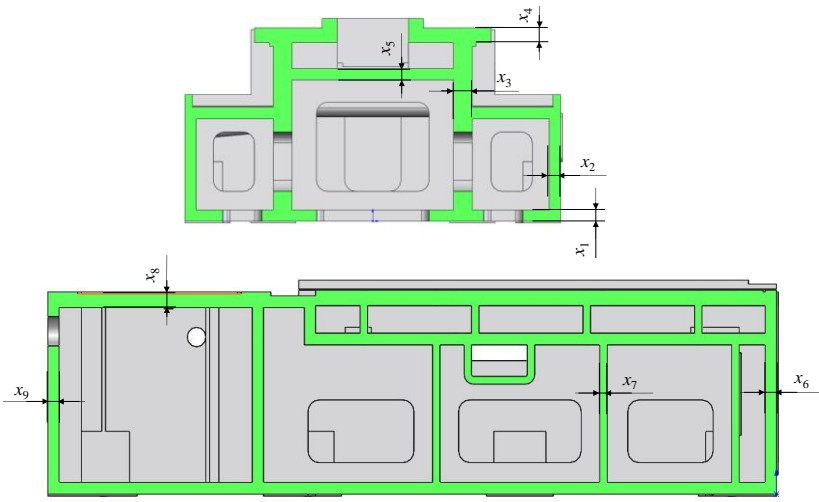

**Figure 8.** Sectional sketch for the design variables.

Owing to the heavy computation time, the data set for the milling machine bed design was fixed; thus, it was not possible to generate 100 different designs of experiments cyclically. To solve this problem, in each of the 100 runs, the points for the data sets ($N$, $N_{add}$, $N_t$) were chosen randomly at respective ratios from the 200 sampling points. The results of the test are listed in Table 4.

**Table 4.** Comparison for the design of milling machine bed.

| Metric [1] | KRG | RBF | SVR | BP | PWS | NPWS | OWS | RLS-EM |
|---|---|---|---|---|---|---|---|---|
| RMSE | 87.86 | 82.15 | 115.53 | 82.97 | 81.88 | 88.61 | 89.82 | **79.87** |
| AAE | 77.79 | **55.75** | 90.81 | 76.73 | 72.24 | 71.60 | 61.44 | 56.88 |
| $R^2$ | 0.82 | 0.82 | 0.85 | 0.82 | 0.84 | 0.86 | 0.85 | **0.87** |

[1] The best error value in each category is shown in bold for ease of comparison.

From Table 4, RLS-EM has the best RMSE and $R^2$ values; further, it has the second-best AAE value. The performance of BP is better than that of KRG, SVR, and other ensembles in AAE; however, the performances of RMSEs of OWS, NPWS, and OWS are better than those of the individual surrogate models. The results reveal that because the linear or nonlinear relationships inside are unknown, when encountering a black-box engineering problem, using an individual surrogate model to approximate the relationship between the design variables and the responses may yield inaccurate results. However, the inaccuracies of each individual surrogate model do not considerably affect the approximate accuracy of RLS-EM; the RLS-EM performs well for the deformation prediction of the milling machine bed. When all the samples are obtained by a time-consuming FEA analysis process, there is no significant increase in the amount of computation caused by the search of the regularization parameter compared to the time-consuming error metrics used in other ensemble methods. Therefore, RLS-EM is an effective engineering problem modeling method that effectively improves the computational efficiency while keeping the modeling accuracy.

## 5. Results

In this work, a new method that combines the advantages of least squares method, the regularization, and the augmentation is developed to construct a better and time-saving ensemble model in the cases that only a small number of sample points are available. The weight factors are calculated by the least squares method with the regularization strategy and the augmentation strategy. The augmentation strategy helps to obtain the augmented samples in the unexplored regions by a sample exploration method. On one hand, it helps to improve the accuracy of the individual surrogate models; on the other hand, the augmentation strategy helps to reduce the collinearity problem caused by the intrinsic properties of KRG and RBF and the approximate prediction values on some densely distributed regions. The regularization strategy with an optimal search method to find the best regularization parameter helps to further reduce the collinearity and avoid the potential overfitting problem.

Six numerical functions and a 9-D CNC milling machine bed deformation prediction problem were used to test the proposed RLS-EM method. Four other ensemble models and KRG, RBF, and SVR were adopted for comparison with RLS-EM. The results show that for the numerical functions, the RLS-EM model can provide satisfactory robustness and accuracy, with better or equivalent levels compared to other ensemble methods, while saving a considerable amount of computational cost. The results of the CNC milling machine bed deformation prediction problem also show that the RLS-EM has a good accuracy and robust performance.

In the future work, the hyperparametric optimization will be studied in the RLS-EM, which will further help improve the accuracy and robustness of the RLS-EM, and RLS-EM-based optimization will be studied too.

**Author Contributions:** Conceptualization, P.Z. and S.Z.; methodology, S.Z.; software, P.Z.; validation, L.Q., G.Y. and J.L.; formal analysis, X.L.; investigation, X.L.; resources, G.Y.; data curation, L.Q.; writing—original draft preparation, P.Z.; writing—review and editing, S.Z.; visualization, L.Q.; supervision, X.L.; project administration, S.Z.; funding acquisition, S.Z.

**Funding:** This research was funded by the National Natural Science Foundation of China, grant number 51675478 and 51875515, the Natural Science Foundation of Zhejiang Province, grant number LY18E050001, and Youth Funds of the State Key Laboratory of Fluid Power and Mechatronic Systems of Zhejiang University, grant number SKLoFP_QN_1702.

**Acknowledgments:** The authors would like to thank Viana for the open SURROGATES Toolbox, and the authors would like to thank the editors for their work, and if there is an opportunity, the authors also want to thank the anonymous reviewers for their constructive comments and suggestions.

**Conflicts of Interest:** The authors declare no conflict of interest.

## Appendix A

*Appendix A.1. Kriging (KRG)*

The basic assumption of KRG is the estimation of the response in the form of:

$$f(x) = p(x) + z(x) \tag{A1}$$

where $f(x)$ is the response value of the function, $p(x)$ is a known polynomial that globally approximates the response, and $z(x)$ is the stochastic component that generates deviations such that the Kriging model interpolates the sampled response data. $z(x)$ has a mean value of zero and covariance as:

$$\text{cov}[x_i, x_j] = \sigma^2 \mathbf{R}(x_i, x_j) \tag{A2}$$

$\mathbf{R}(x_i, x_j)$ is a correlation function between the data points $x_i$ and $x_j$, when choosing Gaussian; it is represented as:

$$R(x_i, x_j) = \exp[-\theta|x_i - x_j|_2], \theta > 0 \tag{A3}$$

Once the correlation function vector has been established, the response can be predicted as:

$$\hat{f}(x) = \hat{\beta} + \mathbf{r}^{\mathrm{T}}(x)\mathbf{R}^{-1}(\mathbf{f} - \mathbf{1}\hat{\beta})$$
$$\hat{\beta} = \frac{1R^{-1}\mathbf{f}}{\mathbf{1}^{\mathrm{T}}R^{-1}\mathbf{1}} \tag{A4}$$

where the matrix $\mathbf{R}^{-1}$ is the inverse of the correlation matrix $\mathbf{R}$ whose elements $R_{ij}$ are computed by (A3), $\mathbf{f}$ is the vector of the sample responses, and $\mathbf{1}$ is an $n \times 1$ vector of ones. $\mathbf{r}(x)$ is calculated by:

$$\mathbf{r}^{\mathrm{T}}(x) = (\text{cov}[x, x_1], \text{cov}[x, x_2], \cdots, \text{cov}[x, x_n]) \tag{A5}$$

Using parameter estimation methods, $\gamma = [\beta, \sigma^2, \theta]$ can be calculated by maximum likelihood estimation, and the detailed derivation of Kriging can be found in [5]. In this work, the MATLAB Kriging toolbox DACE [48] was used.

*Appendix A.2. Radial Basis Function (RBF)*

The radial basis function interpolant has the form of

$$\hat{f}(x) = \sum_{i=1}^{n} \lambda_i \phi(\|x - x_i\|) + p(x) \tag{A6}$$

where $n$ denotes the number of sample points, $\lambda_i$ are the known coefficients to be determined, $p(x)$ is the polynomial item, and $\|x - x_i\|$ represents the Euclidean distance between $x$ and $x_i$. $\phi(.)$ is the Gaussian basis function, which is defined as:

$$\phi(r) = \exp(-\frac{r^2}{2\gamma^2}) \tag{A7}$$

Other forms of the basis functions can be found in [7]. In the present study, we use different $\gamma$ values and polynomial items for different dimensional variables. The unknown parameters $\lambda_i$ and the coefficients of $p(x)$ are obtained as the solution of the linear equations in a matrix system.

$$\begin{pmatrix} \Phi & P \\ P^{\mathrm{T}} & 0 \end{pmatrix} \begin{pmatrix} \lambda \\ c \end{pmatrix} = \begin{pmatrix} F \\ 0 \end{pmatrix} \tag{A8}$$

where $F = [f(x_1), f(x_2), \cdots, f(x_n)]^{\mathrm{T}}$ represent the response values of the $n$ samples.

*Appendix A.3. Support Vector Regression (SVR)*

SVR approximates a linear function $f(x)$ in the following form:

$$f(x) = w^T x + b \tag{A9}$$

where the coefficients $w$ and $b$ are the weight vector and bias term, respectively. This linear function can be constrained to the following optimization problem [10,11,52]. Using $\varepsilon$ as the insensitive loss function, the corresponding SVR, which is called $\varepsilon$-SVR, can be represented as follows:

$$\left| y - (w^T x + b) \right| = \begin{cases} 0, & \left| (w \cdot x_i + b) - y_i \right| < \varepsilon \\ \left| (w \cdot x_i + b) - y_i \right| - \varepsilon, & \left| (w \cdot x_i + b) - y_i \right| \geq \varepsilon \end{cases} \tag{A10}$$

where $\varepsilon$ is a positive constant. The characteristic of this function is that the fitting errors, which are below $\varepsilon$ can be ignored; thus, it has strong anti-noise properties. To measure the degree of deviation from the $\varepsilon$ insensitive band of training samples, two relaxation factors are introduced; thus, the objective function of the SVR optimization is:

$$\min_{w \in R^N, b \in R} \frac{1}{2} \|w\|^2 + C \sum_{i=1}^{n} (\xi_i + \xi_i^*) \tag{A11}$$

Further, the constraint conditions are:

$$\begin{aligned} wx_i + b - y_i &\leq \varepsilon + \xi_i, \quad i = 1, 2, \cdots, n \\ y_i - (wx_i + b) &\leq \varepsilon + \xi_i^*, \quad i = 1, 2, \cdots, n \\ \xi_i, \xi_i^* &\geq 0 \end{aligned} \tag{A12}$$

By introducing a Lagrange function for the optimization problem, the mathematical expression of SVR can be obtained by solving the dual formula:

$$f(x) = \sum_{i=1}^{m} (a_i - a_i^*) K(x_i, x) + b \tag{A13}$$

where $m$ is the number of support vectors, $K(x_i, x)$ is the kernel function, and $a_i$ and $a_i^*$ are the Lagrange multipliers; $b$ is obtained by:

$$\begin{aligned} b &= y_j - \sum_{i=1}^{n} (a_i - a_i^*) K(x_i, x_j) + \varepsilon, a_j \in (0, C) \\ b &= y_k - \sum_{i=1}^{n} (a_i - a_i^*) K(x_i, x_j) - \varepsilon, a_k^* \in (0, C) \end{aligned} \tag{A14}$$

In this paper, the Gaussian kernel was used; it is shown as:

$$k(x_i, x) = \exp\left(-\frac{\|x - x_i\|^2}{\sigma^2}\right) = \exp(-\gamma \|x - x_i\|^2), \gamma > 0 \tag{A15}$$

## Appendix B

*Appendix B.1. Branin-Hoo Function*

$$\begin{aligned} y(x_1, x_2) &= \left( x_2 - \frac{5.1 x_1^2}{4\pi^2} + \frac{5 x_1}{\pi} - 6 \right)^2 + 10 \left( 1 - \frac{1}{8\pi} \right) \cos(x_1) + 10 \\ x_1 &\in [-5, 10], x_2 \in [0, 15] \end{aligned} \tag{A16}$$

*Appendix B.2. Camelback Function*

$$y(x_1, x_2) = (4 - 2.1x_1^2 + \tfrac{x_1^4}{3})x_1^2 + x_1 x_2 + (-4 + 4x_2^2)x_2^2 \tag{A17}$$
$$x_1, x_2 \in [-2, 2]$$

*Appendix B.3. Hartman Functions*

$$y(x) = -\sum_{i=1}^{4} c_i \exp[-\sum_{j=1}^{n} \alpha_{ij}(x_j - p_{ij})^2] \tag{A18}$$

where $x_i \in [0, 1]$. Two types of Hartman functions are provided based on the number of input variables: Hartman-3 with three inputs, and Hartman-6 with six input variables. The parameter $c$ for both is the same vector $[1\ 1.2\ 3\ 3.2]^T$; the other parameters are listed in Tables A1 and A2.

**Table A1.** Parameters used in Hartman function (3-D), $j = 1, 2, 3$.

| $a_{ij}$ | | | $p_{ij}$ | | |
|---|---|---|---|---|---|
| 3.0 | 10 | 30 | 0.3689 | 0.1170 | 0.2673 |
| 0.1 | 10 | 35 | 0.4699 | 0.4387 | 0.7470 |
| 3.0 | 10 | 30 | 0.1091 | 0.8732 | 0.5547 |
| 0.1 | 10 | 35 | 0.03815 | 0.5743 | 0.8828 |

**Table A2.** Parameters used in Hartman function (6-D), $j = 1, 2, \ldots, 6$.

| $a_{ij}$ | | | | | | $p_{ij}$ | | | | | |
|---|---|---|---|---|---|---|---|---|---|---|---|
| 10 | 3.0 | 17.0 | 3.5 | 1.7 | 8.0 | 0.1312 | 0.1696 | 0.5569 | 0.0124 | 0.8283 | 0.5886 |
| 0.05 | 10.0 | 17.0 | 0.1 | 8.0 | 14.0 | 0.2329 | 0.4135 | 0.8307 | 0.3736 | 0.1004 | 0.9991 |
| 3.0 | 3.5 | 1.7 | 10.0 | 17.0 | 8.0 | 0.2348 | 0.1451 | 0.3522 | 0.2883 | 0.3047 | 0.6650 |
| 17.0 | 8.0 | 0.05 | 10.0 | 0.1 | 14.0 | 0.4047 | 0.8828 | 0.8732 | 0.5743 | 0.1091 | 0.0381 |

*Appendix B.4. Extended-Rosenbrock Function*

$$y(x) = \sum_{i=1}^{m-1} [(1 - x_i)^2 + 100(x_{i+1} - x_i^2)^2] \tag{A19}$$
$$x_i \in [-5, 10], i = 1, 2, \cdots, 9$$

*Appendix B.5. Dixion-Price Function*

$$y(x) = (x_1 - 1)^2 + \sum_{i=2}^{m} i[2x_i^2 - x_{i-1}]^2 \tag{A20}$$
$$x_i \in [-10, 10], i = 1, 2, \cdots, m = 12$$

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
