# Peer review of "A Least Squares Ensemble Model Based on Regularization and Augmentation Strategy"

_applsci, doi:10.3390/app9091845_

Round 1

Reviewer 1 Report

Dear authors,

Your article A Least Squares Ensemble Model Based on Regularization and Augmentation Strategy introduces a new approach to numerically simulate higher dimensional functions.

The study consists of an introduction about different single modelling approaches, a section about the new ensemble modelling, the algorithms for combing these results and improving the stability of the numerical computations, a comparison of the outcome of the single computation models to the results of the new averaging approach, and a discussion of the results. The numerical results of the different computation schemes are displayed in a number of diagrams (for 2D functions) and compared to the analytical functions. The approach is then utilized for a deformation calculation of a mechanical specimen. In summary, the RLS-EM model is found to have a better performance than single model approaches. Future work is discussed shortly. An appendix provides information about the functions that are numerically modelled by the novel RLS-GM approach.

The article explores an interesting field within numerical computation and is of interest to the scientific community. It provides an algorithm for an ensemble computation of high dimensional surrogate models interesting within this research field with good methodology and in a sound scientific manner.

My main criticism about the manuscript is the layout of the results. In Tables 1 and 3 it is hard to follow which information / numbers belong to which model. In Table 3 it is unclear what is displayed. It is explained somewhere in the text l 249 but at a quick glance it is quite hard to see what is displayed (mean, standard deviation) and what should be expected. I am not sure that I do understand what mean is taken. I guess it is over the 100 runs mentioned at l 241. The authors should be more precise. It is essential to improve this flaw.

In Figs. 3,4,5 it is unclear what the symbols mean. I guess it is the mean of the different models and the confidence interval. But then there are single dots outside these intervals. Explain.

The English style is up to some typos fine.

Some more specific comments about the manuscript

L 110 fatrors -> factors

L 119 No free index i on the RHS.

L 123 N is not explained. I guess it is the number of sample points.

L 230 is -> are

L 230 fucntion ->function

Table 1 Layout strange. Readability is awkward. It is hard to see which line belongs to the different approaches.

Table 3 Readability awkward, lines between the different functions. What is displayed? -> caption

L 360 helps obtain -> helps to obtain

In total, the study is ready for publication after a revision addressing my comments.

Best regards,

Author Response

We are very grateful to your critical comments and thoughtful suggestions on our manuscript entitled “A Least Squares Ensemble Model Based on Regularization and Augmentation Strategy” (ID: applsci-489874). Those comments are all valuable and very helpful for revising and improving our manuscript, as well as the important guiding significance to our research. Based on these comments and suggestions, we have made careful modifications on the original manuscript. In order to solve the errors in format, spelling, etc., we have tried our best to read our manuscript repeatedly, and we have made some corrections which we hope to meet with approval. The modifications in the manuscript are marked by the "Track Changes". The main corrections in the manuscript and the responses to the reviewer’s comments are as following:

Point 1: My main criticism about the manuscript is the layout of the results. In Tables 1 and 3 it is hard to follow which information / numbers belong to which model. In Table 3 it is unclear what is displayed. It is explained somewhere in the text l 249 but at a quick glance it is quite hard to see what is displayed (mean, standard deviation) and what should be expected. I am not sure that I do understand what mean is taken. I guess it is over the 100 runs mentioned at l 241. The authors should be more precise. It is essential to improve this flaw.

Response 1: We appreciate the reviewer's attention to the flaws of our text, after examining the reviewer’s comments carefully, we must admit that the layout of the two tables mentioned by the reviewer does make people feel confused. In order to make the two tables easier to express our meaning correctly in the manuscript, we made the following modifications:

1)      There is a numbering error, in L 244, “Table 1. Surrogate models setup details” should be “Table 2. Surrogate models setup details”.

2)      In the review comments, the reviewer has pointed out that “In Tables 2 and 3 it is hard to follow which information / numbers belong to which model.” In order to clearly describe the one-to-one correspondence among the number of variables, the model and its details, we made some modifications to the layout of Table 2 and 3. At L 244, in Table 2, we have set the column labelled “ndv” in the form of “Top centred”, which makes the number of variable for each numerical case, the corresponding KRG model and its details on the same line. We have indented the column labelled “Details”, making each model name of different dimensional variables appear on the same line as its corresponding representations.

3)      In order to facilitate the understanding of the table description, at L 235 of the manuscript, we have added some textual expressions, that is, after the sentences “Table 2 lists the setup …… based on different variable dimensions and nonlinearities.” in the manuscript, the sentences “Each individual model has significant differences between variables with different dimensions and different degrees of nonlinearity, e.g., for the low-dimensional variables such as variable with numbers 2 and 4, constant regression can satisfy the accuracy requirements of a KRG model, while the high-dimensional variables require quadratic regression to obtain a more accurate model. Similarly, the kernel parameters and regularization parameters of different dimensional variables with different degrees of nonlinearity are different for the SVR model. Thus, the KRG, RBF and SVR model setting information for different dimensional variables are listed in detail as shown in Table 2.” were added.

4)      In Table 3, we have set the column labelled “Function” and “Metric” both in the form of “Top centred”, which makes the function name, the first Metric indicator, and the mean value of the indicator of different models on the same line.

5)      The reviewer has pointed out that “In Table 3 it is unclear what is displayed. It is explained somewhere in the text L249 but at a quick glance it is quite hard to see what is displayed (mean, standard deviation) and what should be expected. I am not sure that I do understand what mean is taken. I guess it is over the 100 runs mentioned at L 241. The authors should be more precise. It is essential to improve this flaw.” Considering the reviewer’s comments and suggestions, we have made the following modifications of the manuscript: at L 256, the sentences “After 100 runs were executed, the mean and standard deviation of the RMSE, AAE and R2 metrics for the numerical examples are shown in Table 3. For each metric of the numerical examples, the values to the left of the symbol “/” are the mean of the different models, and the values below are the standard deviations corresponding to the models. For the RMSE, AAE metrics, the smaller mean values indicate the better model accuracy, and the smaller standard deviation values show the better robustness. The R2 metric whose mean value is closer to 1 and the standard deviation is smaller indicates the more accurate and more robust model” were added. We are very grateful to the reviewer for these comments, which helps us to more easily discuss the models at different metrics, make our presentations easier to understand, and make the modified tables more clearly show the content of the expression

Point 2: In Figs. 3,4,5 it is unclear what the symbols mean. I guess it is the mean of the different models and the confidence interval. But then there are single dots outside these intervals. Explain.

Response 2: We are grateful to the reviewer's comments, In Figure 3,4 and 5, we use the boxplot to provide a graphical depiction of how the value of each metric varies over the range of training samples used. A boxplot is a standardized way of displaying the distribution of data based on a five number summary (Minimum, first quartile (Q1), Median, third quartile (Q3), and Maximum). It can tell us about the outliers and what their values are, it can also tell us if the data is symmetrical, how tightly the data is grouped, and if and how the data is skewed. we use a graphical representation in Figure 1(in the word file) to explain the different parts of a boxplot.

The box is defined by lines at the lower quartile (1/4, Q1), median (1/2) and upper quartile (3/4, Q3) of the data. Lines extending above and upper each box (“whiskers”) indicate the spread for the rest of the data out of the quartiles definition, the outliers are represented by the red plus signs “+” above/below the whiskers. Consistent with the reviewer’s comments, the boxplot is actually a description of the confidence interval, and the purpose of using the boxplot is also to provide the visualization toolkit of the model accuracy and robustness. The “single dots outside these intervals” are the outliers in which we use the 5-95 percentile rules when we create the plots. In the manuscript, we use the toolkit “GraphPad Prism” to create the plots in our manuscript.

Point 3: Some more specific comments about the manuscript.

Response 3: We are very sorry for our incorrect writing and the strange layout errors, and we are very grateful to the reviewer and the editors for their valuable comments on the content of the manuscript, the spelling of the words and the layout of the tables. As suggested by the reviewer, we have made some modifications.

1)      L 110 “fatrors” has been removed, we are grateful to the editors for their modification.

2)    L 119 “No free index I on the RHS:” Eq. (3) was corrected in the word file.

3)      L 123 “N is not explained”: “N is the number of sample points” was added at L 123.

4)     L 230 “is->are, function-> function”. “Details of each test fucntion is given in Appendix B.” were corrected as A description of each test is given in Appendix B” at L 229, we are grateful to the editors for their modifications.

5)  “Table 2 Layout strange. Readability is awkward. Table 3 Readability awkward, lines between the different functions. What is displayed.” As the reviewer suggested, we have made some modifications of the layout for Table 2 and 3 at L 244 and L 280. We have also added some text narratives to make the tables more readable with the help of sentences. 

6) L 360 “helps obtain -> helps to obtain”. In L 369, “The augmentation strategy helps obtain the augmented samples” were corrected as “The augmentation strategy helps to obtain the augmented samples”

We tried our best to improve the manuscript and made some changes in the manuscript. These changes will not influence the content and framework of the manuscript, and we use the “Track Changes” function in the manuscript.

We appreciate for the Editors and Reviewers’ warm work earnestly, and hope that the correction will meet with approval.

Once again, thank you very much for your comments and suggestions.

Wish you all the best!

Sincerely yours,

Peng Zhang                                

Reviewer 2 Report

The authors aim to develop RLS-EM. The methodologies they suggested are quite reliable and have extensive potential to use. But moderate English changes required.

For example:

A space is needed before round bracket, "(".

, on the other hand,   => ; on the other hand,

What does the CNC stand for?

There is a numbering error. "3. Results" should be "5. Results."

Author Response

Dear Editors and Reviewers:

We are very grateful to your critical comments and thoughtful suggestions on our manuscript entitled “A Least Squares Ensemble Model Based on Regularization and Augmentation Strategy” (ID: applsci-489874). Those comments are all valuable and very helpful for revising and improving our manuscript, as well as the important guiding significance to our research. Based on these comments and suggestions, we have made careful modifications on the original manuscript. In order to solve the errors in format, spelling, etc., we have tried our best to read our manuscript repeatedly, and we have made some corrections which we hope to meet with approval. The modifications in the manuscript are marked by the "Track Changes". The main corrections in the manuscript and the responses to the reviewer’s comments are as following:

Point 1: A space is needed before round bracket, "(", “,on the other hand,”   => “; on the other hand,”, There is a numbering error. "3. Results" should be "5. Results.".

Response 1: We are very grateful to the reviewer for the comments on the format, spelling, etc., and we are very sorry for our incorrect writing on format, spelling, numbering. We have read the manuscript carefully and tried our best to correct the errors, including the formatting issues, the punctuations, the spelling errors, etc. The changes made are as follows:

1)  In L 16, a space was added before “(RLS-EM)”.
2)  In L 23, “CNC” was modified as “computer numerical control (CNC)”.
3)  In L 28, “showed” was corrected as “shows”.
4)  In L 109, L 122, L 141, L413, the space before “where” was removed.
5)   In L 164, “Collinearity” was modified as “collinearity”.
6)  In L 170, a space was added before                                                .
7)  In L 174, The variables representing the vectors and the matrices in
Algorithm 1 were corrected in the bold form. X-> X ,S-> S , Xlhs -> Xlhs ,
Xadd -> Xadd.
8)   In L 177, L 186, “Xadd” was bolded as “Xadd”.
9)  In L 208, “l = 1, r= q” was corrected as “l = min= 1, r= max= q”.
10)  In L 270, L 271, and L 272, the three blank lines caused by adding other sentences were removed.
11)  In L 282, a space was added before “5”.

12)  In L 324 and L325, “CNC” was added before “milling machine”.

13)  In L 352 and L 353, “was” and “were” were corrected as “is” and “are”.

Point 2: What does the CNC stand for?

 Response 2: We appreciate the reviewer's attention to the issue that the abbreviation “CNC” does not have a corresponding full name. CNC stands for “Computer Numerical  Control”, and in L 23, “CNC” was modified as “computer numerical control (CNC)”. In L 324 and L325, “CNC” was added before “milling machine”.

Point 3: There is a numbering error. "3. Results" should be "5. Results."

 Response 3: We are grateful to the reviewer's comments and we are very sorry for our incorrect writing about the numbering error. In L 397, the "3. Results" was corrected as "5. Results."

We tried our best to improve the manuscript and made some changes in the manuscript. These changes will not influence the content and framework of the manuscript, and we use the “Track Changes” function in the manuscript.

We appreciate for the Editors and Reviewers’ warm work earnestly, and hope that the correction will meet with approval.

Once again, thank you very much for your comments and suggestions.

Wish you all the best!

Sincerely yours,

Peng Zhang
